# Sparse World Models: Visual World Modeling with Sparse Representations

## Abstract

World models promise efficient prediction, imagination, and planning by operating in a compact latent space, yet prevailing approaches inherit *dense, entangled* visual features from large pretrained encoders. Such latents conflate unrelated factors and contain redundant dimensions, undermining intervention fidelity, inflating planning cost, and reducing robustness to distribution shifts. We propose **Sparse World Models (SWMs)**, which learn and plan *entirely in a sparse feature space*. SWMs obtain selectively active codes by training a sparse autoencoder (SAE) to translate dense vision embeddings into an overcomplete but *sparse* vocabulary, and then use these codes for state estimation, dynamics learning, and action optimization. By aligning units to meaningful factors, SWMs enable targeted interventions and attribution, and shrink the optimization search space. We further introduce an evaluation suite that probes feature capacity and links sparsity to planning outcomes. Across studies, sparse representations reduce polysemanticity and maintain planning performance while offering better efficiency and interpretability.

## 1 Introduction

World models (WMs) aim to capture environment dynamics in a compact latent space, enabling agents to predict, imagine, and plan without operating directly on pixels (Ha & Schmidhuber, 2018; Hafner et al., 2019a; 2020; 2023). However, in practice, learning such latent representations poses a major challenge: uncertainty and perceptual noise often obscure the true dynamics necessary for effective planning. Recent methods, thus, leverage large, pretrained vision backbones to extract visual features, which helps improve performance but yields *dense and entangled* representations (Ha & Schmidhuber, 2018; Hafner et al., 2019a; 2020; 2023; Zhou et al., 2024). These representations suffer from two key limitations: (1) Entanglement: A single latent unit may respond to multiple unrelated factors (e.g., both object rotation and background texture), making it difficult to interpret activations or intervene on specific variables; (2) Redundancy: Many latent dimensions encode overlapping or irrelevant information, inflating dimensionality without adding task-relevant content.

As a result, dense representations limit world models in three recurring ways. *(i) Intervention fidelity and attribution:* When latent units encode multiple unrelated factors (e.g., object pose and background texture), interventions on one variable may unintentionally alter others. This makes it difficult to isolate causal effects or attribute actions to specific visual inputs. *(ii) Planning cost:* Redundant features diffuse task-relevant information across correlated dimensions, inflating the search space. Planners need to optimize over more axes, increasing both runtime and variance–especially with wider latents and longer horizons. *(iii) Robustness to distribution shifts:* Entanglement mixes task and nuisance variables, so minor changes (e.g., in lighting, viewpoint, or layout) can activate irrelevant latents, corrupt transition inputs, and lead to cascading errors over time. Feature-space world models that bypass pixel reconstruction (e.g., planning directly on visual features) reduce reconstruction burden yet still operate on *dense* representations and therefore inherit these limitations.

Rather than processing all visual information indiscriminately, humans focus on a few task-relevant cues and ignore the rest. Similarly, world models should rely on *sparse* visual representations—where only a small subset of units activate per scene, each aligned with a meaningful and interpretable factor. This selective encoding preserves essential predictive signals while suppressing nuisance variation. As a result, sparse representations directly mitigate the limitations above: they

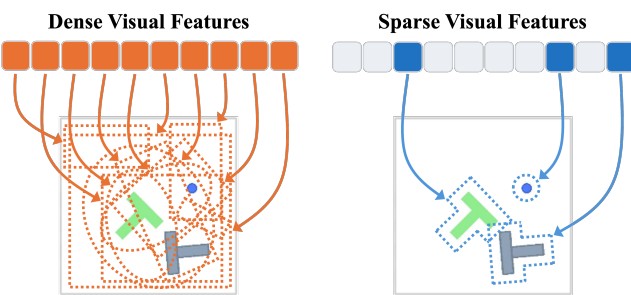

Figure 1: Comparison of dense and sparse visual features. Sparse features yield more monosemantic units and lower redundancy, making downstream planning and attribution more effective.

enhance intervention fidelity, reduce planning complexity, and improve robustness to minor distribution shifts. This raises the following questions:

*Can world models learn and plan entirely within a sparse feature space? If so, how can such a space be identified, and how can learning and planning be effectively carried out within it?*

We propose **Sparse World Models (SWMs)** that operate *entirely in sparse space*. As illustrated in Fig. 1, dense visual features are often redundant and entangled, whereas sparse codes exhibit *selective activation* and tend toward *monosemanticity* (approximately one dimension per factor). Enforcing sparsity in visual representations reshapes the model's latent state so that (i) factor-aligned units improve the fidelity of targeted interventions and attribution; (ii) optimizers search a narrower space, reducing planning cost; and (iii) task-irrelevant features stay inactive under routine changes in viewpoint, lighting, or texture, improving robustness.

To obtain such representations, we enforce sparsity via a *sparse autoencoder* (SAE). An SAE learns an overcomplete code in which only a small subset of units activate for a given input, encouraging *selective*, often human-interpretable units and reducing superposition. Prior studies show that SAEs can reveal monosemantic directions and support causal interventions in large language models, and sparsify pre-trained vision embeddings while preserving semantics (Cunningham et al., 2023; Templeton et al., 2024; Wen et al., 2025). This provides a practical path to *interpretable, selectively active* latents without modifying the upstream perception model. Accordingly, for our SWMs, we train an SAE to translate dense visual features from a vision encoder into a sparse, structured vocabulary, keeping the salient "words" (factors) and discarding task-irrelevant ones.

**Contributions.** (i) We show that SAE-derived visual representations reduce polysemanticity in WMs, yielding monosemantic features aligned with meaningful environment factors. (ii) We demonstrate that sparse features can replace dense embeddings without degrading *planning performance*, while improving efficiency and interpretability. (iii) We introduce an evaluation suite, including feature probes and attribution overlap, that connects sparsity to planning outcomes.

## 2 RELATED WORKS

### 2.1 VISUAL REPRESENTATION LEARNING IN WORLD MODELS

Early world models learned visual representations *from scratch* with reconstruction objectives. The "World Models" framework encoded images with a VAE and rolled forward with an RNN to support imagination-based control (Ha & Schmidhuber, 2018). Latent dynamics models such as PlaNet and Dreamer families advanced this idea by learning compact stochastic latents from pixels and using imagined rollouts for policy optimization (Hafner et al., 2019b;a; 2020; 2023).

A recent trend replaces training vision encoders end-to-end with *frozen, pre-trained* backbones to obtain stronger features from image observations. In robotics and RL, such features improve sample efficiency and generalization by providing semantically rich visual states before learning dynamics or policies (Nair et al., 2022; Linsley et al., 2024; He et al., 2022; Radford et al., 2021). In world-model settings, DINO-WM shows that patch-level ViT features (DINOv2) enable learning dynamics

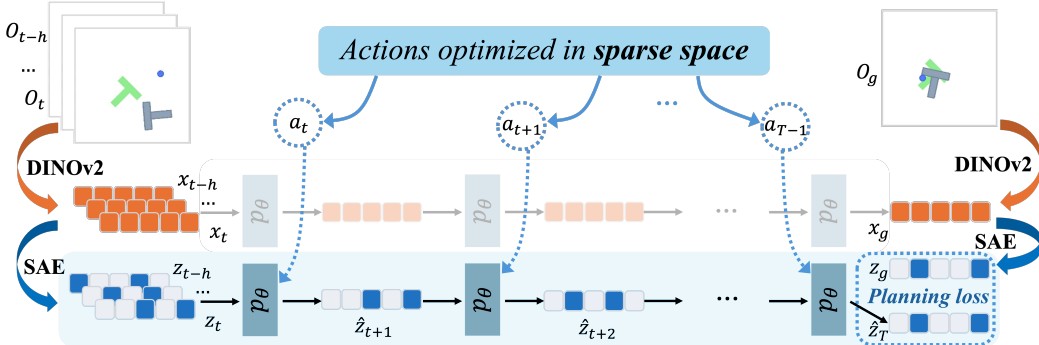

Figure 2: Overview of our Sparse World Models. We operate entirely in the sparse latent space: during training, the transition model $p_\theta$ is fit on sparse sequences; during planning, actions are optimized via rollouts in the same space.

and planning directly in feature space, sidestepping pixel reconstruction altogether and preserving spatial structure useful for control (Zhou et al., 2024). However, even with powerful pre-training, the resulting representations are typically *dense* and *polysemantic*: individual dimensions respond to multiple unrelated factors, and many channels are redundant, which inflates planning cost and complicates interpretation.

## 2.2 SPARSE REPRESENTATIONS

Sparse representations activate only a small subset of latent units per input, yielding *selectively active* features that are easier to attribute and probe than dense embeddings. Classic sparse coding showed that optimizing for sparsity on natural images recovers localized, edge-like primitives (Olshausen & Field, 1996). Neural variants enforce sparsity either by keeping only the top-$k$ activations or via competition: $k$-Sparse Autoencoders retain the $k$ largest hidden responses per example (Makhzani & Frey, 2013), while Winner-Take-All (WTA) autoencoders impose lifetime and spatial sparsity through competitive inhibition (Makhzani & Frey, 2015).

Recent *sparse autoencoders* (SAEs) trained on foundation-model activations extract *monosemantic* units aligned with human-interpretable concepts, supporting attribution, unit-level ablations, and targeted activation edits. In language models, SAEs decompose residual streams into interpretable features and scale to millions of latents with low dead-unit rates (Cunningham et al., 2023; Bricken et al., 2023; Templeton et al., 2024). In vision/vision–language models, hierarchical SAEs recover multi-granularity concept banks from CLIP while maintaining high reconstruction fidelity under substantial sparsity (Zaigrajew et al., 2025). For adaptive inference budgets, Contrastive Sparse Representation (CSR) sparsifies frozen embeddings into a selectively active high-dimensional code, preserving semantic quality while allowing sparsity-controlled compute (Wen et al., 2025). These results suggest that SAEs can map dense features into a *sparse space* without retraining the encoder, thereby narrowing the planner's effective search space and clarifying which units drive decisions; at test time, sparsity provides a principled knob for the compute–accuracy trade-off.

## 3 SPARSE WORLD MODELS

We propose operating world models *entirely in sparse space*, mitigating the redundancy and polysemanticity commonly observed in dense features. Here, we describe (i) how we obtain sparse visual representations with sparse autoencoders (SAEs), (ii) a decoder-free world model trained purely in the sparse space, and (iii) test-time planning with model predictive control (MPC) in that space.

### 3.1 SPARSE AUTOENCODERS (SAEs)

Let $o_t$ be an observation (i.e., an image). A frozen vision encoder $f_{\text{vis}} : \mathcal{I} \to \mathbb{R}^{P \times d}$ maps $o_t$ to dense per-patch features $x_t = f_{\text{vis}}(o_t) = [x_{t,1}, \ldots, x_{t,P}]$, with $x_{t,p} \in \mathbb{R}^d$. An SAE with encoder $W_{\text{enc}}$ :

$\mathbb{R}^d \to \mathbb{R}^{h_z}$, decoder $W_{\mathrm{dec}} : \mathbb{R}^{h_z} \to \mathbb{R}^d$, and sparsity operator TopK produces per-patch sparse codes $z_{t,p} = \mathrm{TopK}\big(W_{\mathrm{enc}}(x_{t,p})\big) \in \mathbb{R}^{h_z}$. Stacking over patches yields $z_t = [\, z_{t,1}, \ldots, z_{t,P}\,] \in \mathbb{R}^{P \times h_z}$. The decoder reconstructs dense features per patch, $\hat{x}_{t,p} = W_{\mathrm{dec}}\, z_{t,p}$, forming $\hat{x}_t \in \mathbb{R}^{P \times d}$.

Let $W_{\mathrm{enc}} \in \mathbb{R}^{h_z \times d}$, $W_{\mathrm{dec}} \in \mathbb{R}^{d \times h_z}$, and biases $b_{\mathrm{enc}}, b_{\mathrm{pre}}$ (a learned pre-mean). With ReLU $\sigma_+(u) = \max(0, u)$ and $\mathrm{TopK}(\cdot)$ selecting the $k$ largest elements,

$$z_t = \sigma_+\Big( \mathrm{TopK}\big(W_{\mathrm{enc}}(v_t - b_{\mathrm{pre}}) + b_{\mathrm{enc}}\big)\Big), \tag{1}$$

$$\hat{v}_t = W_{\mathrm{dec}} z_t + b_{\mathrm{pre}}, \tag{2}$$

and the reconstruction loss at sparsity $k$ is

$$\mathcal{L}_{\mathrm{recon}}^{(k)} = \big\| v_t - \hat{v}_t \big\|_2^2. \tag{3}$$

**Dead-latent mitigation.** To reduce dead latents (Templeton et al., 2024), we adopt tied initialization ($W_{\mathrm{enc}} \leftarrow W_{\mathrm{dec}}^\top$ at start), unit-norm column renormalization of $W_{\mathrm{dec}}$, and an auxiliary "AuxK" loss that asks a small set of currently *inactive* units to explain the residual during training. We denote the auxiliary regularizer by $\mathcal{L}_{\mathrm{aux}}$ and include it when training the SAE (Gao et al., 2024; Wen et al., 2025).

**Unsupervised sparse contrastive learning.** Beyond pure reconstruction, we shape $z$ with a non-negative contrastive objective. For a minibatch $\{z_i\}_{i=1}^B$ (each $z_i \geq 0$ elementwise due to $\sigma_+$), the loss is

$$\mathcal{L}_{\mathrm{ncl}} = -\frac{1}{B} \sum_{i=1}^B \log \frac{\exp\big(z_i^\top z_i^+\big)}{\exp\big(z_i^\top z_i^+\big) + \sum_{j \neq i}^B \exp\big(z_i^\top z_j\big)}. \tag{4}$$

We obtain $z_i$ and its positive $z_i^+$ from two stochastic augmentations of the *same* frame, and treat all other codes in the batch as negatives, requiring no labels and thus making the objective fully unsupervised. This encourages discriminative, identifiable non-negative features while keeping sparsity control via $k$ (Wen et al., 2025).

**Overall SAE objective.** Our SAE is trained with

$$\mathcal{L}_{\mathrm{SAE}} = \mathcal{L}_{\mathrm{recon}}^{(k)} + \mathcal{L}_{\mathrm{aux}} + \mathcal{L}_{\mathrm{ncl}}. \tag{5}$$

## 3.2 SPARSE WORLD MODELS (SWMS)

We operate the world model entirely in the sparse latent space produced above. At time $t$, given an observation $o_t$ and action $a_t$, the model has three parts:

$$\underbrace{z_t = \mathrm{TopK}\big(W_{\mathrm{enc}}(X_t)\big),}_{\text{Observation model}} \qquad \underbrace{\hat{z}_{t+1} = p_\theta\big(z_{t-h:t}, a_{t-h:t}\big),}_{\text{Transition model}} \qquad \underbrace{\hat{x}_t = W_{\mathrm{dec}}\, z_t \,\big[\text{and } \hat{o}_t = q(\hat{x}_t)\big].}_{\text{Decoder (optional)}}$$

Here $h \geq 1$ is the history length, $p_\theta$ is a parametric dynamics model in the sparse space. The decoder is used only for feature/pixel visualization (e.g., inspecting $\hat{x}_t$ or, if an external image decoder $q$ is available, $\hat{o}_t = q(\hat{x}_t)$) and is not required for training $p_\theta$ or for planning; all learning and control operate on the sparse codes $\{z_t\}$.

### 3.2.1 OBSERVATION MODEL

We follow the *feature-space world model* paradigm, representing observations with a frozen, task-agnostic vision backbone. We use DINOv2 due to its strong spatial priors and robust transfer to detection, segmentation, and depth tasks. (Oquab et al., 2023) Given $o_t$, the backbone yields patch embeddings $X_t = f_{\mathrm{vis}}(o_t) \in \mathbb{R}^{P \times d}$.

To mitigate the polysemanticity and redundancy of dense features, we attach a *spare autoencoder* (SAE) on top of the frozen backbone and take the resulting sparse codes as the observation state. Applying the encoder $W_{\mathrm{enc}}$ and sparsity operator TopK row-wise over patches gives

$$z_t = \mathrm{TopK}\big(W_{\mathrm{enc}}(X_t)\big) \in \mathbb{R}^{P \times h_z},$$

---

**Algorithm 1** Train SWM Transition $p_\theta$

---

**Require:** Sequential data $\{(o_t, a_t)\}_{t=1}^N$; frozen $f_{\text{vis}}, W_{\text{enc}}, \text{TopK}$; history $h$; action encoder $\phi$;
    regime $\texttt{mode} \in \{\texttt{full}, \texttt{active}\}$
**Ensure:** Trained transition $p_\theta$
  1: **Pre-encode:** $z_t \leftarrow \text{TopK}\big(W_{\text{enc}}(f_{\text{vis}}(o_t))\big)$ for all $t$
  2: Initialize $\theta$
  3: **repeat**
  4:     Sample a center index $t$ with $t \geq h$ and $t+1 \leq N$
  5:     $u_z \leftarrow z_{t-h:t}; \quad u_a \leftarrow \phi(a_{t-h:t})$
  6:     $\hat{z}_{t+1} \leftarrow p_\theta(u_z, u_a)$                                            (Eq. equation 6)
  7:     $z^\star \leftarrow z_{t+1}$
  8:     **if** $\texttt{mode} = \texttt{full}$:   $\mathcal{L} \leftarrow \|\hat{z}_{t+1} - z^\star\|^2$
  9:     **else**  (active):   $\mathcal{L} \leftarrow \text{LossActive}(\hat{z}_{t+1}, z^\star)$
 10:     Update $\theta$ to minimize $\mathcal{L}$
 11: **until** convergence

---

which preserves the backbone's task-agnostic spatial structure while translating dense features into a *selectively active*, more *monosemantic* latent on which our dynamics and planner operate (Fig. 2). For clarity, our observation mapping is

$$\text{Obs}(o_t) \; = \; \text{TopK}\big(W_{\text{enc}}(f_{\text{vis}}(o_t))\big).$$

### 3.2.2 Transition Model

*Goal.* Learn a dynamics map from a short history of sparse latents and actions to the next sparse latent.

We write $z_{u:v} = [z_u, \dots, z_v]$ and $a_{u:v} = [a_u, \dots, a_v]$, with $z_t \in \mathbb{R}^{P \times h_z}$. Given history length $h$, a sequence of actions $a_{t-h:t}$, and an action encoder $\phi$ that embeds actions into the model input space, a parametric dynamics model $p_\theta$ maps a *sequence of sparse latents and actions* to the next sparse latent:

$$\hat{z}_{t+1} \; = \; p_\theta\big(z_{t-h:t}, \phi(a_{t-h:t})\big), \qquad z_{t+1} \; = \; \text{TopK}\big(W_{\text{enc}}(f_{\text{vis}}(o_{t+1}))\big). \tag{6}$$

With teacher forcing, the single-step prediction loss is

$$\mathcal{L}_{\text{pred}} \; = \; \big\| \hat{z}_{t+1} - z_{t+1} \big\|^2. \tag{7}$$

**Two supervision regimes.** Our SAE learns an *overcomplete* code ($h_z \gg d$) but enforces sparsity via the $\text{TopK}$ operator: a dense pre-activation $u_t = W_{\text{enc}}(X_t)$ is mapped to a $k$-sparse code $z_t = \text{TopK}(u_t)$. This yields two natural targets for training the transition model:

**1. Full-sparse.** Predict the entire next code $\hat{z}_{t+1} \in \mathbb{R}^{P \times h_z}$ and regress to $z_{t+1}$:

$$\mathcal{L}_{\text{full}} = \|\hat{z}_{t+1} - z_{t+1}\|^2.$$

This maximizes *coverage*: every channel is trained to carry forward dynamics. It is the most faithful to the original representation and best when broad consistency of the feature space matters (e.g., for decoding/analysis across many channels). Since the SAE's dictionary is overcomplete, this regime encourages the model to propagate all represented factors that survive $\text{TopK}$, offering broad coverage of the learned basis.

**2. Active-set (Top-$k$ only).** Supervise only the entries that are active in the ground-truth code at $t+1$:

$$\mathcal{L}_{\text{active}} = \big\| M_{t+1} \odot (\hat{z}_{t+1} - z_{t+1}) \big\|^2, \quad M_{t+1} = \mathbb{1}[z_{t+1} \neq 0] \text{ (Top-}k\text{ mask)}.$$

This maximizes *focus*: it directs capacity to the control-salient subset that the planner will actually use, aligns with the SAE's sparsity pattern, and reduces compute. It is preferred when planning quality and efficiency are paramount and we want to avoid spending capacity on rarely used channels.

*When to use which.* Full-sparse is a good default for analysis, image reconstruction, or when we expect task-relevant signal to spread across many channels. Active-set is preferred for control-heavy regimes where a small set of features drives behavior; it mirrors the planner's reliance on Top-$k$ coordinates and concentrates learning on those directions.

Training process is summarized in Algorithm 1.

### 3.2.3 DECODER (FOR INTERPRETABILITY)

For visualization only, we map sparse codes back to dense features via the SAE decoder $\hat{X}_t = W_{\text{dec}} z_t$. If desired, a separately trained image decoder $q_\psi$ (kept fixed thereafter) maps dense features to pixels, $\hat{o}_t = q_\psi(X_t)$, with reconstruction loss $\mathcal{L}_{\text{img}}(\psi) = \| \hat{o}_t - o_t \|_2^2$. At analysis time we may also render $q_\psi(\hat{X}_t)$ to visualize sparse rollouts. This component is not used to train $p_\theta$ and is not used during planning.

### 3.3 PLANNING WITH SWMs

At test time, given a current observation $o_0$ and a goal observation $o_g$, we encode them into sparse latents via the observation pipeline,

$$z_0 = \text{TopK}\big(W_{\text{enc}}(f_{\text{vis}}(o_0))\big), \qquad z_g = \text{TopK}\big(W_{\text{enc}}(f_{\text{vis}}(o_g))\big).$$

We perform MPC with the cross-entropy method (CEM) over action sequences $a_{0:T-1}$, rolling out the learned dynamics equation 6 in the sparse space. Let $z_{-h+1:0}$ denote the last $h$ observed latents; we initialize

$$\hat{z}_{-h+1:0} = z_{-h+1:0}, \qquad \hat{z}_{t+1} = p_\theta\big(\hat{z}_{t-h+1:t}, \phi(a_{t-h+1:t})\big) \text{ for } t = 0, \ldots, T-1.$$

The planning cost is the terminal latent distance to the goal:

$$\mathcal{C}(a_{0:T-1}) = \big\| \hat{z}_T - z_g \big\|^2. \tag{8}$$

CEM iteratively samples candidate action sequences, refits a proposal to elite samples under equation 8, and executes the first action in a receding-horizon loop.

## 4 EXPERIMENTS

We test a single claim: *planning entirely in a sparse feature space preserves control performance while concentrating task signal and reducing compute.* Accordingly, we evaluate along two axes. *A1 — Representation & dynamics.* Do sparse visual features improve spatial selectivity without degrading the transition model's next-state attribution? Sparsity should sharpen *where* the model attends while keeping *what* the dynamics relies on comparably grounded, showing that a sparse substrate is at least as attributionally faithful as a dense one. *A2 — Planning quality & efficiency.* Do sparse features sustain high success while converting compute to performance more efficiently? We assess both the end-point quality (success rate) and the compute–performance trajectory (success versus time/operations), expecting sparse predictors to reach high-performance thresholds with fewer operations.

Please see Appendix A.1 and Appendix A.2 for Experiment Setup.

### 4.1 SAE PRETRAINING

We pretrain the SAE on *fixed* DINOv2 ViT-S/14 patch tokens (384-d) extracted from $196 \times 196$ images; all setup details are summarized in Table 5. Unless otherwise noted, we use Top-$k{=}128$ sparsity and sweep $k \in \{32, 64, 256\}$ in ablations. For efficiency, we train for 10 epochs on a $\sim 10\%$ subsample of the data. The objective is reconstruction-dominated, and all remaining hyperparameters follow the public defaults.[1]

### 4.2 FEATURE PROBING

If sparse representations emphasize *control-relevant* cues, those variables (goal, action) should be *more linearly accessible* than in dense embeddings. Because our planner operates entirely in feature space, greater linear accessibility should translate into easier prediction, lower optimization cost, and improved interpretability.

---

[1] https://github.com/neilwen987/CSR_Adaptive_Rep

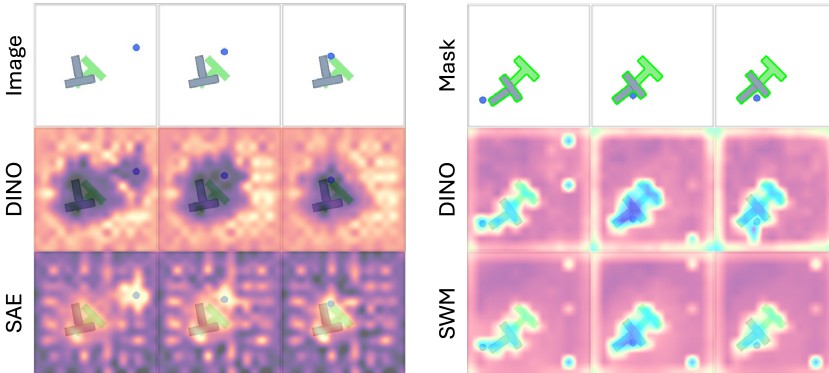

Figure 3: **Visual vs. dynamics attribution.** *Left:* visual feature heat map highlighting spatial regions/features with high attribution in visual features; DINO denotes DINOv2. *Right:* dynamics heat map showing channels/patches most influential for next-state prediction in the learned transition model; DINO denotes DINO-WM.

Following Qi et al. (2024), we freeze the visual encoder and train lightweight linear heads on single frames to predict two labels: (i) *goal* (identity/region) and (ii) *action* (the executed control, discretized). We compare dense DINOv2 features against sparse SAE features trained on the same backbone. Probes use identical train/val splits, cross-entropy, and a fixed training budget, with no augmentation or feature finetuning. Higher is better (accuracy).

**Discovery.** Across Maze, PushT, and Wall, SAE features consistently improve *action* decoding and match or exceed DINOv2 on *goal*. The largest gains occur on PushT, where fine-grained control matters, supporting the view that sparsity concentrates control salience into a small set of readout-friendly directions. This aligns with our frequency analysis (Sec. A.8): SWM repeatedly reuses a compact subset of channels, whereas dense features distribute signal more broadly. Overall, sparsity yields representations that are both *more selective* (easier to decode) and *more efficient* for planning.

| | Maze | | PushT | | Wall | |
|---|---|---|---|---|---|---|
| **Feature** | goal | action | goal | action | goal | action |
| DINOv2 | **0.58** | 0.35 | 0.68 | 0.52 | 0.78 | 0.65 |
| SAE | 0.57 | **0.37** | **0.74** | **0.62** | **0.82** | **0.68** |

Table 1: **Control-oriented feature probing (accuracy).** Linear probes on frozen features. SAE yields consistent gains on *action* decoding and improves *goal* decoding on PushT and Wall, indicating that sparsity concentrates control-relevant information.

### 4.3 REPRESENTATION AND DYNAMICS ATTRIBUTION

From Tab. 2, SAE improves *spatial selectivity* of visual features over DINOv2 (IoU 0.16 vs. 0.00), indicating that sparse codes localize task-relevant regions that dense features largely miss. At the same time, *dynamics attribution* remains unchanged between models (MC 0.048 for both DINO-WM and SWM), showing that moving to a sparse substrate does not erode what the transition model relies on. Qualitatively, the left heat maps in Fig. 3 are sharper and more localized for SAE, whereas the right panels show comparable patterns for next-state influence across DINO-WM and SWM. Taken together, these results support the intended tradeoff: sparsity concentrates where the model looks, while preserving what the dynamics depends on.

### 4.4 PLANNING.

From Tab. 3, SWM (Top-$k$) matches the best result on *Maze* (1.00) and is competitive on *Wall* (0.92 vs. 0.96 for DINO-WM), while landing second on the manipulation-heavy *PushT* (0.86 vs. 0.90). Notably, it does so using only *128* active sparse channels per step (one third of the 384-

| Visual features | | Dynamics | |
|---|---|---|---|
| Method | IoU ↑ | Method | MC ↑ |
| DINOv2 | 0.00 | DINO-WM | 0.048 |
| SAE | 0.16 | SWM (full) | 0.048 |

Table 2: Visual feature quality (IoU) and dynamics quality (MC).

| Method | Maze | PushT | Wall |
|---|---|---|---|
| DreamerV3 | **1.00** | 0.30 | **1.00** |
| TDMPC | 0.00 | 0.00 | 0.00 |
| DINO-WM | 0.98 | **0.90** | 0.96 |
| SWM (Top-$k$) | **1.00** | 0.86 | 0.92 |

Table 3: **Planning success rate** (higher is better) on *Maze*, *PushT*, and *Wall*. SWM uses a *Top-k* predictor with *128-D* inputs (active sparse coordinates), whereas DINO-WM uses *384-D* dense DINO features. Best in **bold**, second-best underlined.

D dense input used by DINO-WM), indicating that a focused, sparsity-aligned predictor retains the task signal needed for planning with far fewer features. DreamerV3 performs strongly on navigation (*Maze*, *Wall*) but struggles on *PushT* (0.30), and TDMPC fails on this suite due to lack of reward. Overall, these results support our claim that concentrating learning and inference on the *active* SAE coordinates yields efficient, high-success planning, narrowing the gap to dense-feature models while reducing representational overhead.

| | CEM | | | GD | | | MPC | | |
|---|---|---|---|---|---|---|---|---|---|
| Method | Maze | PushT | Wall | Maze | PushT | Wall | Maze | PushT | Wall |
| DINO-WM | 0.80 | **0.86** | 0.74 | **0.22** | 0.28 | *N/A* | 0.98 | **0.90** | **0.96** |
| SWM (full) | **1.00** | 0.74 | **0.78** | 0.16 | 0.32 | 0.60 | 0.96 | 0.76 | 0.88 |
| SWM (Top-$k$) | **1.00** | 0.70 | 0.70 | **0.22** | **0.46** | **0.62** | **1.00** | 0.86 | 0.92 |

Table 4: **Planning success rate** (higher is better) across three planners: CEM, direct gradient-based planning (GD) in latent space, and MPC with CEM.

From Tab. 4, across planners, **MPC** is consistently strongest: SWM (Top-$k$) reaches **1.00** on *Maze* and 0.92 on *Wall*, while DINO-WM leads *PushT* at **0.90**. Under **CEM** (open-loop shooting), both SWM variants hit **1.00** on *Maze*; SWM (full) edges out others on *Wall* (0.78), whereas DINO-WM remains best on *PushT* (0.86). The clearest separation appears with **GD** (gradient action optimization): SWM (Top-$k$) achieves the best scores on all available tasks (0.22 / **0.46** / **0.62**), suggesting that concentrating supervision and inference on the *active* sparse coordinates yields a smoother, more informative planning landscape for gradient-based methods. Overall, focusing on Top-$k$ features trades a small amount of *PushT* performance versus DINO-WM under MPC, but delivers robust wins for GD and competitive CEM results, supporting our claim that sparse, focused latents are especially well-suited to gradient-driven planning while remaining strong under stochastic and receding-horizon schemes.

## 4.5 SR vs. Computation Cost

In Fig. 4, we evaluate *planning efficiency* on *Maze* by plotting success rate (SR) against *cumulative computation* during planning. Each curve traces how SR grows as compute accrues, exposing not just the final SR but also the *speed* at which high-performance thresholds are reached. We compare DINO-WM (purple) and SWM (Top-$k$, blue): steeper slopes and shorter time-to-threshold (e.g., 80%/90% SR) indicate better compute-to-performance conversion.

To quantify compute, we use token operations (`token_ops`): the cumulative count of token-level operations issued during planning. This proxy scales with FLOPs and correlates with wall-clock time while avoiding profiler noise and hardware-specific effects. Since transformer cost grows

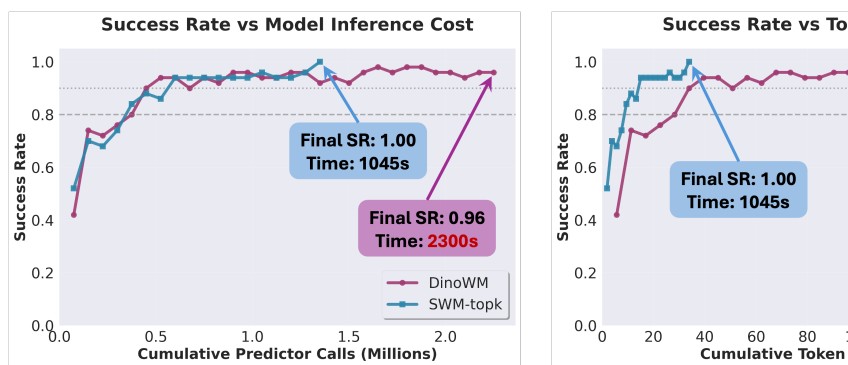

Figure 4: **SR vs. computation on *Maze*.** *Left:* SR vs. cumulative time. *Right:* SR vs. cumulative `token_ops`. DINO-WM (purple) vs. SWM (Top-$k$, blue). Dashed lines mark 80%/90% SR.

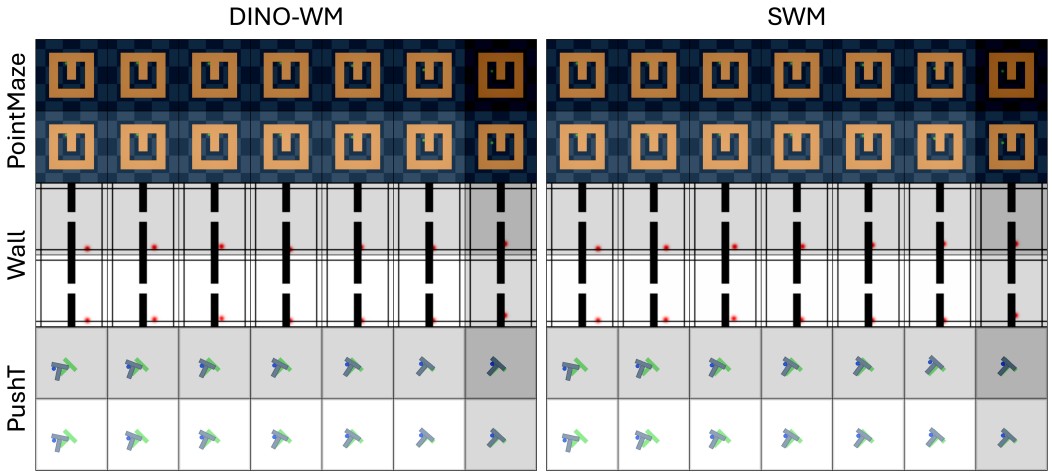

Figure 5: **Planning visualizations on *Maze* (top), *Wall* (middle), and *PushT* (bottom).** *Left:* DINO-WM. *Right:* SWM (full-sparse).

roughly with $d \times S$ (latent width $d$, number of patches $S$), SWM-Top$k$ (128-D) consumes far fewer token operations per step than DINO-WM (384-D), yielding faster SR gains per unit compute.

### 4.6 PLANNING VISUALIZATION (COMBINED).

Figure 5 presents side-by-side planning rollouts on *Maze* (top two rows), *Wall* (middle two rows), and *PushT* (bottom two rows). The **left column** shows DINO-WM and the **right column** shows SWM (full-sparse). Within each environment block, the **upper (shaded) row** depicts the *executed observations* obtained by running the planned actions in the environment, and the **lower row** shows the model's *imagined observations* along that plan. Across *Maze* and *Wall*, both models recover goal-directed paths.

## 5 CONCLUSION

Sparse World Models operate entirely in a sparse feature space, preserving control performance while concentrating task-relevant signal and reducing compute. Across various tasks, sparse features improve attribution selectivity, maintain predicted-state quality, and achieve strong success with better success–compute tradeoffs.

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

## A APPENDIX

### A.1 EXPERIMENTAL SETUP

We evaluate on three image-based control environments that stress complementary skills:

- **PushT (manipulation).** A T-shaped block must be pushed to a target pose. Success requires contact reasoning and precise action sequencing. This setting probes whether sparse features capture *action*-relevant cues for manipulation.
- **PointMaze (navigation).** A point agent must reach a goal region through a maze. This emphasizes long-horizon wayfinding with minimal contact dynamics, isolating planning over geometry.
- **Wall (navigation with bottleneck).** The agent must pass through a narrow gap between walls to reach the goal. This stresses spatial selectivity (identifying the passage) and robust rollout around obstacles.

**Observations and actions.** Unless noted, observations are RGB frames resized to $224 \times 224$ and encoded by a frozen DINOv2 ViT-S/14 into patch tokens; SWM replaces dense tokens with sparse SAE codes (Top-$k$ by default). Actions are environment-native controls (e.g., planar motions or end-effector deltas) applied at a fixed frameskip.

**Planners and budgets.** We report results with three planners: stochastic shooting with CEM, gradient-based action optimization (GD) in feature space, and receding-horizon MPC. Planner horizons, samples/iterations, and termination conditions follow the standard protocol used across baselines; all methods share the same episode budgets.

**Metrics.** Primary metrics are *success rate (SR)* for control, *SR vs. time*/`token_ops` for efficiency, and attribution/rollout diagnostics: IoU and MC Jiang et al. (2024) for representation/dynamics grounding, LPIPS/SSIM (context only) for pixel fidelity, and linear probing accuracy for control-factor accessibility.

### A.2 DATASET LABELING

We construct *control-oriented* labels that are decodable from a *single* frame to probe whether representations linearly expose task factors.

| Aspect | Setting |
|---|---|
| Backbone features | DINOv2 ViT-S/14 patch tokens (384-d) |
| Input resolution | $196 \times 196$ |
| Training epochs | 10 |
| Data fraction | $\sim 10\%$ of full dataset |
| Sparsity (default / sweep) | Top-$k$=128 / $k \in \{32, 64, 256\}$ |
| Latent size (overcomplete) | 1536 ($4 \times 384$) |
| Objective | Reconstruction-dominated; others at defaults |

Table 5: SAE pretraining configuration used throughout our experiments.

For navigation environments (PointMaze, Wall), we extract spatial relationship features between the agent and target locations. Since stored trajectory data lacks explicit target position information, we infer target regions from final frame positions where agents successfully reach goals. Labels are generated based on relative position encoding: (i) $\mathbf{dx}, \mathbf{dy}$: Spatial displacement between agent and target in world coordinates; (ii) $\theta$: Angular displacement toward target direction; (iii) **Region Classification**: 8-class discretization based on spatial quadrants and angular bins. For manipulation (*PushT*), we adopt the standard object/pose and discretized action labels (compass directions + stop) used in large-scale robotic representation studies.

Full labeling details and conventions closely follow Jiang et al. (2024).

### A.3 GRADIENT-BASED ATTRIBUTION ANALYSIS

To further understand how sparse representations affect attention patterns and attribution quality, we analyze feature attributions using gradient-based visualization methods. We implement both standard Grad-CAM and Mask-guided Token-CAM (MT-CAM) to compare how dense DINO-WM features versus sparse SWM features attend to task-relevant regions.

#### A.3.1 ATTRIBUTION METHOD

For each model, we compute attention heatmaps by backpropagating gradients through the visual encoder to identify which spatial regions most strongly influence the model's predictions. For DINO-WM, we apply Grad-CAM to the final transformer block of the DINOv2 backbone operating on dense 384-dimensional features. For SWM, we extend the analysis to the sparse autoencoder's top-$k$ active features, focusing on the most salient sparse dimensions.

We evaluate attribution quality using the Mask Coverage (MC) metric (need reference here), which measures the intersection-over-union (IoU) between the binarized attention map and ground-truth object masks:

$$\text{MC} = \frac{|\text{Attention\_Binary} \cap \text{GT\_Mask}|}{|\text{Attention\_Binary} \cup \text{GT\_Mask}|}. \tag{9}$$

### A.4 FEATURE PROBING ABLATIONS

**Setup.** We freeze the visual encoders and train lightweight probes on PushT to predict two task-relevant predicates: *goal* and *action*. Unless noted, probes are trained on a fixed subset of image patches and evaluated on a held-out set; higher is better. We report two ablations: (i) varying sparsity dimension $k$ at a fixed training set size (10M patches), and (ii) varying the training set size with fixed $k$=128. The DINOv2 row probes dense DINOv2 features directly (no SAE), while the other rows probe SWM's sparse codes.

**Observations from Tab. 6.** Relative to DINOv2, sparse features improve alignment with task factors: at $k$=128 we see +0.06 (goal) and +0.08 (action), and at $k$=256 the gains reach +0.07 / +0.10. Very small $k$=32 roughly matches dense probing on *goal* but lags on *action*, suggesting insufficient capacity for contact/interaction cues.

**Observations from Tab. 7.** (1) *Data helps*: 10M $\rightarrow$ 50M yields modest further gains, especially for *action* (+0.0281 over 10M). (2) *Sweet spot*: 10–50M patches are sufficient; "ALL" underperforms

Table 6: Probing on PushT with a fixed 10M-patch training set while varying $k$.

| Method / Dim. | Goal | Action |
|---|---|---|
| DINOv2 (dense 384) | 0.68 | 0.52 |
| SWM-$k$=32 | 0.67 | 0.52 |
| SWM-$k$=64 | 0.71 | 0.56 |
| SWM-$k$=128 | 0.74 | 0.60 |
| SWM-$k$=256 | **0.75** | **0.62** |

Table 7: Probing on PushT with fixed $k$=128 while varying the number of training patches.

| Train size | Goal | Action |
|---|---|---|
| DINOv2 (dense 384) | 0.68 | 0.52 |
| 1M | 0.68 | 0.53 |
| 10M | 0.74 | 0.60 |
| 50M | 0.74 | **0.62** |
| ALL | 0.66 | 0.52 |

10–50M, consistent with label/temporal noise or domain heterogeneity diluting probe signal. (3) *Takeaway*: $k$=128 with 10–50M patches is a strong default, achieving most of the improvement while keeping representation compact.

**Summary.** Sparse codes yield more linearly separable factors than dense DINOv2 features, particularly for *action*-related cues. Scaling $k$ improves probing up to $k\approx128$–256 with diminishing returns, and moderate pretraining sets (10–50M patches) are sufficient. We adopt **k=128** with **50M** patches on PushT as the default, as it offers the best efficiency–accuracy trade-off.

## A.5 SWM IMPLEMENTATION DETAILS

**Component architectures.** The predictor is a ViT operating on sparse features from an SAE. It stacks depth-6 Transformer blocks (LN–MHSA–FFN) with 16 heads and 2048-d MLPs, returning token-wise outputs in the same embedding dimension. We embed the action (and proprioception, if available) with a lightweight *MLP* and *concatenate* the resulting vectors to each patch token before the transformer. The image decoder is a VQ-VAE–style upsampler.

| Component | Default config |
|---|---|
| Predictor | `depth`=6, `heads`=16, `mlp_dim`=2048, `dropout`=0.1 |
| Action encoder | MLP: Linear(`action_dim` $\to$ 64)$\to$GELU$\to$Linear(64 $\to$ 10) |
| Proprio encoder | MLP: Linear(`proprio_dim` $\to$ 64)$\to$GELU$\to$Linear(64 $\to$ 10) |
| Image decoder | `channel`=384, `n_res_block`=4, `n_res_channel`=128, `n_embed`=2048 |

Table 8: SWM component architectures.

**Training hyperparameters.** The visual encoder is *frozen*. We use Adam/AdamW with distinct learning rates for the encoder, predictor, optional decoder, and the action/proprio encoders. The decoder is optional: when enabled, it reconstructs sparse codes back to DINOv2 features (and can be paired with an RGB head if desired). Other implementation details follow Zhou et al. (2024).

## A.6 OPEN-LOOP ROLLOUTS.

Figure 6 visualizes open-loop predictions on *PointMaze*, *Wall*, and *PushT*. Across all environments, SWM closely tracks GT over long horizons. The qualitative parity between rows highlights our core strength: by operating entirely in a sparse feature space, SWM maintains sharp, consistent dynamics with minimal drift, yielding rollouts that visually match GT sequences while preserving task-critical structure.

| Parameter | Value | Description |
|---|---|---|
| training.batch_size | 32 | Global batch size |
| training.encoder_lr | $1\times10^{-6}$ | Encoder learning rate (Adam) |
| training.decoder_lr | $3\times10^{-4}$ | Decoder learning rate (Adam) |
| training.predictor_lr | $5\times10^{-4}$ | Predictor learning rate (AdamW) |
| training.action_encoder_lr | $5\times10^{-4}$ | Action/proprio encoders LR (AdamW) |
| img_size | 224 | Input image size |
| frameskip | 5 | Temporal stride between steps |
| action_emb_dim | 10 | Action embedding dimension |
| proprio_emb_dim | 10 | Proprioception embedding dimension |
| num_hist | 3 | History frames used by predictor |
| num_pred | 1 | Future frames predicted |

Table 9: SWM training hyperparameters.

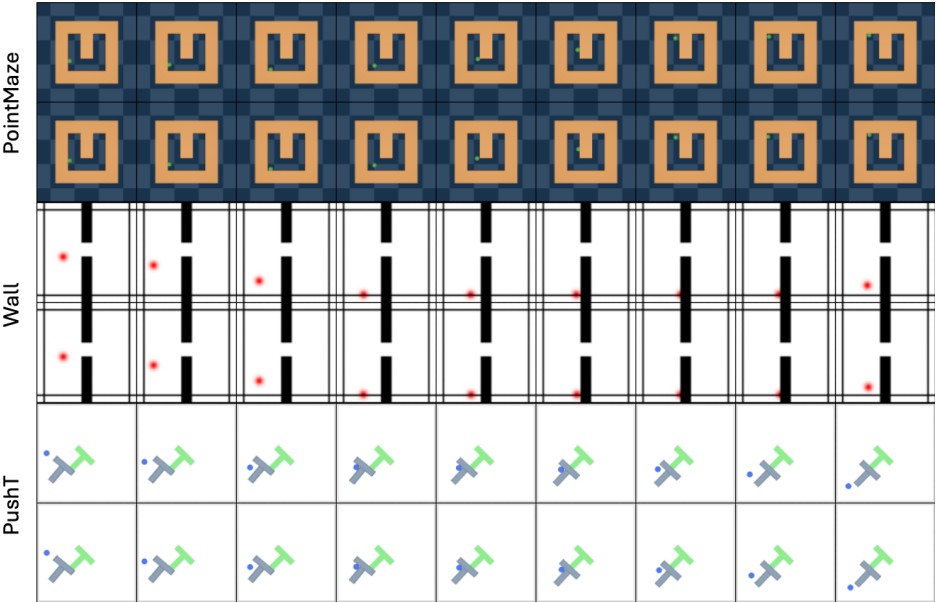

Figure 6: Open-loop rollouts on *PointMaze*, *Wall*, and *PushT*. For each environment, the **top row** shows ground-truth (GT) frames and the **bottom row** shows SWM rollouts.

| Method | LPIPS ↓ | SSIM ↑ |
|---|---|---|
| R3M | 0.045 | 0.956 |
| ResNet | 0.063 | 0.950 |
| AVDC | 0.046 | 0.959 |
| DINO-WM | **0.007** | **0.985** |
| SWM | 0.022 | 0.964 |

Table 10: **Perceptual fidelity on *PushT*.** Lower is better for LPIPS; higher is better for SSIM. Results for baselines are cited from Zhou et al. (2024).

## A.7    IMAGE RECONSTRUCTION QUALITY

DINO-WM attains the best pixel-level fidelity (lowest LPIPS, highest SSIM), which is expected for a method optimized around dense feature reconstruction. **SWM** is competitive, clearly outperforming R3M/ResNet/AVDC on LPIPS, while trailing DINO-WM on both metrics.

Since our goal is to operate *entirely in sparse feature space* for planning; the image decoder is used only for visualization. To minimize compute, we *adapt* the DINO-WM image decoder and perform

only light finetuning as in Sec. 3.2.3, rather than training an image decoder from scratch. *This choice substantially reduces training cost and keeps the focus on our core contribution: planning entirely in sparse feature space.* We expect stronger LPIPS/SSIM if we train a dedicated image decoder end-to-end with SWM; however, our results (Sec. 4.2, Fig. 6) show that SWM already concentrates control-relevant structure and produces stable rollouts that matter more for planning than maximizing dense perceptual scores.

### A.8 FEATURE FREQUENCY ANALYSIS

We quantify how often the same feature channels are important for both the current observation and the one-step prediction. At each planning step, we compute per-patch features in each model's native space: SWM uses DINOv2→SAE sparse codes (1536-D), while DINO-WM uses DINOv2 features (384-D). For each patch, we select the top-$k$ channels by absolute activation for the input and for the predicted next state ($k{=}128$), take their intersection by channel index, and accumulate counts over time and patches to form a histogram. Axes: DINO-WM—x: channel index 0–383, y: overlap frequency; SWM—x: channel index 0–1535, y: overlap frequency.

**Result.** SWM yields a peaked histogram (few high-frequency channels), indicating focused, reusable features; DINO-WM is flatter with many high-frequency channels, indicating broader, more redundant usage (Fig. 7).

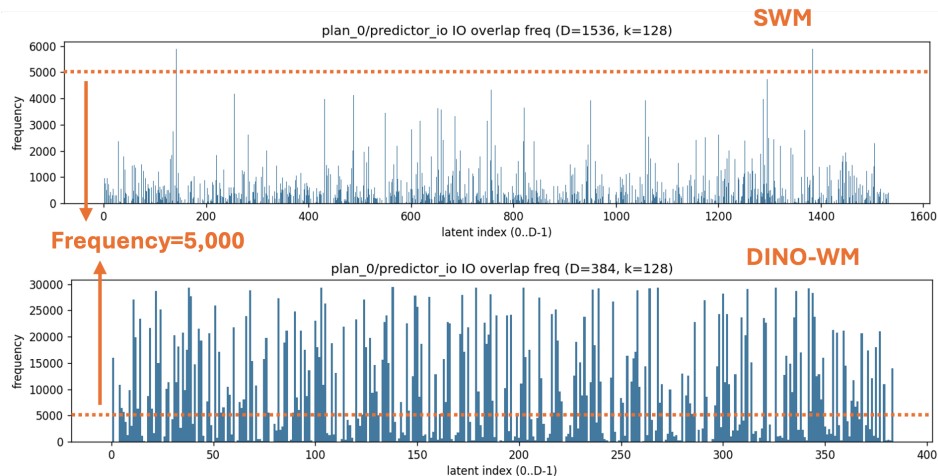

Figure 7: Frequency-overlap histograms of input/prediction top-$k$ co-occurrence in each model's native feature space on PushT.

