# OpenReview forum: "Sparse World Models: Visual World Modeling with Sparse Representations"
_ICLR.cc/2026/Conference — Submitted to ICLR 2026_

### Official Review · Reviewer_Qqco · 2025-10-29

**Soundness:** 2
**Presentation:** 2
**Contribution:** 2
**Rating:** 2
**Confidence:** 4

**Summary:**

The authors propose Sparse World Models (SWM), which make use of a Sparse Autoencoder to disentangle semantically and visually monosemantic features from the dense embeddings of pretrained vision models (like Dino). The authors then investigate whether sparse feature spaces are beneficial for planning. They train an action conditioned latent dynamics model to predict future sparse features from the current sparse features and an action. These sparse features sometimes slightly decrease planning performance, sometimes increase it, but evidence is inconclusive if it actually improves performance in general.

**Strengths:**

* Thorough evaluation of an interesting idea on several tasks.
* Reasonable hypothesis that sparsity would help planning
* Might have potential for speeding up planning algorithms themselves, but it's a little unclear how this works.

**Weaknesses:**

* While the sparse representations do not decrease performance substantially, there doesn't seem to be much evidence for them improving the performance for downstream planning tasks.
* The added SAE pretraining that one needs to do between selecting an encoder and performing planning is probably not worth it given that planning in the SAEs latent space doesn't reliably improve performance (perhaps except for the GD method, but this method has the weakest performance overall).
* The benefit of using SAEs is that they improve interpretability, but the authors do not attempt to interpret the features that the SAEs learn here. This would justify using them, if interpretability is indeed the goal. However, if it is, one could just plan in the dense space and train SAEs for interpretability separately.


If the authors can showcase the benefits of planning in a sparse feature space better, and explain how planning is sped up, I would be willing to increase my score.

**Questions:**

* Action decoding setup is unclear. Is the decoder trained to decode what action the planner executed? This seems circular, since the planner operates in the representation space of the SAE (e.g. minimizes goal distance in the learned sparse representation space).
* How can SWM have lower token_ops count because of its sparsity? In my understanding, the SAE representation is up-projected, and then sparsified. But importantly, the dimensionality of the SAE representation is the up-projected size, not k. I'd appreciate it if the authors could address this and explain how exactly using the SAE representations lead to speed-ups and lower token_ops.
* If TDMPC gets 0 success rate because it doesn't get rewards, why is it included in the comparison? Is this comparison actually informative?

---

> ### Author Response · Authors · 2025-11-27
>
> ***W1:***
> We agree that on tasks like PushT and Wall, sparse representations do not always yield higher raw
> success rates. However, those results are obtained while using only **one-third of the latent
> dimension** (128 vs. 384 in DINO-WM). When we move to a more challenging deformable
> environment such as Rope and use the **full sparse latents** (SAE features built on top of the same
> dense encoder), SWM **significantly outperforms** the dense model:
>
> | Model                 | Rope (CD↓) |
> |-----------------------|------------|
> | DINO-WM               | 0.41       |
> | SWM (full-nodecoder)  | **0.24**   |
> | SWM (full-withdecoder)| **0.32**   |
> | SWM (Top-k, k=128)    | 0.42       |
>
> In addition, SWM is competitive not only against dense baselines but also against other sparse world
> models. Compared to SI-WM (Sparse Imagination; https://arxiv.org/pdf/2506.01392), whose 60–70% token
> drop corresponds to a similar compute reduction as our 128/384 (≈33%) sparsity level, SWM
> (Top-k = 128) achieves the strongest performance on PointMaze, PushT, and Rope under comparable
> compute budgets:
>
> | Method              | Point Maze (SR↑) | PushT (SR↑) | Rope (CD↓) |
> |---------------------|------------------|-------------|------------|
> | SI-WM (drop 60%)    | 83%              | 47%         | 70.0       |
> | SI-WM (drop 70%)    | 70%              | 28%         | 53.3       |
> | SWM (Top-k, k=128)  | **100%**         | **86%**     | **42.2**   |
>
> Finally, sparse latents offer clear structural advantages: Sec. 4.2 shows improved linear
> accessibility of task factors; Sec. 4.3 shows stronger object-centric selectivity (IoU 0.00→0.16);
> Sec. 4.5 shows faster success-per-compute; and App. A.4 reports consistent trends across sparsity
> and data scales.
>
> ***W2:***
> We see the reviewer’s point about the additional SAE pretraining step, but our results indicate that
> the trade-off is worth it.
> The SAE pretraining cost is actually **very small**—on a single H100 GPU, SAE pretraining finishes in
> under 10 minutes. This is because:
>
> 1. We train the SAE on only ~10% of the data for 10 epochs,
> whereas the world model is trained on (approximately) the full dataset for 100 epochs. In total,
> SAE pretraining requires only a tiny fraction of the compute used for world-model training. This
> corresponds to roughly **1%**
> as many optimization steps.
>
> 2. Moreover, SAE
> training uses **single-step image features**, whereas world-model training requires multi-step
> history unrolling, transition prediction, and backpropagation through the dynamics horizon, making
> each WM update far more expensive than an SAE update.
>
> 3. Finally, the SAE itself is a **shallow,
> lightweight encoder–decoder**, while the world model is a **Transformer-based architecture** with
> significantly higher compute per forward/backward pass.
>
> Taken together, SAE pretraining adds only
> a negligible fraction of the total compute while enabling the structural and efficiency benefits of
> sparse latents.

---

> ### Author Response · Authors · 2025-11-27
>
> ***W3:***
> Thank you for raising this point. In our framework, SAEs do more than provide interpretability—they
> serve a **functional role** in the world model. Enforcing sparsity in the latent space reduces the
> planner’s operating dimension (384 → k) and yields a more structured representation for recursive
> rollouts.
>
> Plus, a post-hoc SAE would not alter the dense features actually used by the dynamics or the
> planner, and the surrogate would effectively be a **different** model that cannot faithfully explain the
> behavior of the world model [1].
>
> To evaluate interpretability and intervention fidelity, we added an intervention experiment. For
> paired observations, we compute Δstate (δs) and Δlatent (δh), fit a linear map δŝ = W δh, and then
> apply an intended displacement δs* by modifying the latent (h_cf = h + δh*). We then check
> (i) whether δŝ matches δs* on held-out pairs, and (ii) how much the decoded image changes overall.
>
> **Probe Metrics**
>
> | Model    | Representation | Train MSE | Train R² | Val MSE | Val R² |
> |----------|----------------|-----------|----------|---------|--------|
> | DINO-WM  | Dense          | 60.5      | –14.7    | 69.8    | –16.3  |
> | SWM      | Dense          | 10.7      | –1.77    | 14.8    | –2.67  |
> | SWM      | SAE (sparse)   | **2.00**  | **+0.48**| **5.64**| **–0.39** |
>
> **Intervention Diagnostics** (δs* = [0.1, 0, 0, 0])
>
> | Model / Rep.    | \|δŝ − δs*\| | L2(x₀, x_cf) | Cosine(x₀, x_cf) |
> |-----------------|--------------|--------------|------------------|
> | DINO-WM / Dense | ≈ 0          | 0.97         | 0.999988         |
> | SWM / SAE       | ≈ 0          | **0.60**     | **0.999996**     |
>
> **Analysis**
>
> Dense features fail to produce a meaningful δh → δs mapping (large negative R²), so the edited
> latent direction is not well defined, and interventions cause broad, unintended changes in the image.
> In contrast, SWM’s sparse SAE latents yield much lower MSE and positive train R², indicating that
> the displacement direction is far more linearly recoverable. As a result, applying the same δs*
> produces **substantially smaller image drift** with sparse latents, showing that the change is more
> localized and aligned with the intended factor.
>
> **Conclusion**
>
> This demonstrates that sparse latents are not only more interpretable but also more **causally
> aligned** with controllable state changes. While we do not claim perfect “position-only” edits
> everywhere in the image, the experiment provides clear evidence that SWM achieves **higher
> intervention fidelity** than dense DINO features and that SAE pretraining enables a more functional
> and controllable planning latent space than a post-hoc SAE could provide.
>
> *[1] Rudin, C. Stop explaining black box machine learning models for high stakes decisions and use interpretable models instead. Nat Mach Intell 1, 206–215 (2019). https://doi.org/10.1038/s42256-019-0048-x*

---

> ### Author Response · Authors · 2025-11-27
>
> ***Q1:***
> There is **no** action decoder or planner involved during probing. As described in Sec. 4.2 and App. A.2–A.4, action probing is a **diagnostic linear evaluation task** performed purely on the *offline dataset*, not on planned rollouts. The labels are the environment’s ground-truth actions between frames $t$ and $t{+}1$, exactly following the protocol of [2], and the probe sees only the single image at time $t$. The probe is trained on frozen features, entirely independent of both the planner and the SWM transition model.
>
> Thus, the probing task measures **how linearly accessible the true environment action is from the representation**, not how well the planner’s chosen action can be decoded. The planner never supervises the probe, and the probe is never used during planning. The evaluation is therefore non-circular and serves solely as a representation-quality diagnostic, consistent with standard practice in representation learning for control.
>
> ***Q2:***
> The key point is that the SAE’s overcomplete latent (the \(h_z\)-dimensional code) is **not** the
> representation used by the planner.
>
> Although the SAE produces an \(h_z\)-dimensional vector,
> the transition model and planner operate **only on the Top-\(k\)** active coordinates (Eq. 6 uses the
> sparsified \(z_t\), not the dense pre-activation). Thus, SWM plans in a **k-dimensional latent space**
> (128 by default), while DINO-WM operates in the full **384-dimensional** DINO space. Because
> transformer cost scales roughly linearly with latent width, reducing 384 → 128 yields ~3× fewer
> token-level operations per rollout step, which accumulates significantly over recursive planning
> (N samples × T horizon). This explains the faster success-per-compute curves in Sec. 4.5.
>
> We will clarify this distinction in the revision to avoid confusion between SAE’s internal latent
> dimension and the planner’s actual operating dimensionality.
>
> ***Q3:***
> Thank you for pointing out. We include TDMPC because it is a widely used, strong model-based control baseline, and DINO-WM also reports it under the same reward-free evaluation protocol. In this setting, TDMPC is expected to fail because it requires dense reward signals during training, whereas the feature-space planning setting we evaluate (shared across our work and DINO-WM) provides only observations and actions. The comparison is therefore informative in the same sense as in DINO-WM: it situates SWM relative to both (i) reward-free feature-space planners (DINO-WM, SWM) and (ii) classical reward-driven MPC methods (TDMPC), highlighting that the latter category cannot operate in this regime. We will clarify this motivation in the text.
>
>
> *[2] Jiang, Guangqi, et al. "Robots pre-train robots: Manipulation-centric robotic representation from large-scale robot datasets." arXiv preprint arXiv:2410.22325 (2024). https://arxiv.org/pdf/2410.22325*

---

> > ### Comment · Reviewer_Qqco · 2025-11-27
> >
> > Thanks for your response and your clarifications. I'm still not convinced about the planning in the lower-dimensional top-k space. If you construct latent embeddings only consisting of the active features after the top-k operation, you end up with a strange representation to do planning over. If latent feature 1 is active at time t but not active at time t+1, the vector component in the compressed representation will be occupied by a feature representing something else at t+1. That's why you need the whole sparsified 384-dimensional space, otherwise the representation just tells you the magnitude of a couple of different features, but there is no way of knowing what these features actually correspond to because they can switch position in the vector (depending on what features are active). Even if this ends up working in practice in your experiments, it seems unprincipled and unlikely to work for many environments (e.g. environments where the active features might change a lot between frames).

---

> ### Author Response · Authors · 2025-11-27
>
> Thank you for your follow-up question. We have considered this issue carefully, which is why we include both SWM-full and SWM-Top$k$ in our experiments. As you mentioned, the top-$k$ active features can vary across timesteps, which is exactly why we did not train an image decoder for SWM-Top$k$. However, downstream planning still works well.
>
> This shows an important point: **planning and reconstruction have very different requirements**. Reconstruction needs all visual factors to be preserved, while planning only needs the small set of features that encode controllable geometry (e.g., agent position, obstacles, openings). The good performance of SWM-Top$k$ indicates that sparse features capture the task-relevant information needed for MPC, even if they are not sufficient for pixel-level decoding.
>
> So the inability to reconstruct does not imply an unstable or inconsistent planning space—it simply reflects that planning can work with a much smaller, more focused representation, and the strong performance of SWM-Top$k$ shows this is not a coincidence but a consistent pattern across tasks.
>
> As for scenarios like "environments where the active features might change a lot between frames", we can simply increase $k$ to raise the overlap of active features across timesteps. In the extreme case, we can use all sparse features, i.e., SWM-full, as reported in the paper.

---

### Official Review · Reviewer_GGi3 · 2025-10-30

**Soundness:** 2
**Presentation:** 2
**Contribution:** 2
**Rating:** 4
**Confidence:** 4

**Summary:**

This paper addresses the challenges of inefficient planning, poor interpretability, and a lack of robustness inherent in existing visual World Models, which stem from their reliance on dense, entangled, and redundant visual representations. The paper introduces Sparse World Models (SWMs), a framework that employs a Sparse Autoencoder (SAE) to transform dense features from pretrained encoders (e.g., DINOv2) into an overcomplete yet highly sparse set of activations. Critically, SWMs conduct all subsequent dynamics learning and online planning entirely within this sparse feature space. Experimental results demonstrate that this approach not only maintains (and in some cases surpasses) planning success rates comparable to dense-feature baselines but also significantly reduces feature polysemanticity, enhances attribution accuracy, and substantially improves computational efficiency, accomplishing planning tasks with lower computational overhead and in less time.

**Strengths:**

The paper presents a clear motivation and well-defined problem statement. The methodology and experimental setup are described with clarity, and the overall presentation is concise and easy to follow.

**Weaknesses:**

1. The proposed method primarily combines existing sparse autoencoder techniques with established world model architectures, resulting in an incremental contribution rather than a fundamentally new approach. A major concern is that the paper does not convincingly justify the novelty or benefits of integrating sparse representations with existing world models. Table 2 shows no clear performance advantage of SWM over DINO-WM, and Figure 3 indicates that both DINO (without sparse representations) and SWM capture similar dynamics heat map, casting doubt on the necessity and effectiveness of the sparse representations.

2. The paper lacks a thorough discussion and empirical comparison with prior work on robustness under distribution shifts and task-relevant representation learning, particularly recent bisimulation-based approaches [1, 2]. Moreover, the claim that sparse representations improve robustness to distribution shift is not supported by targeted experiments. To substantiate this claim, the authors should include controlled experiments with explicit distribution shifts (e.g., background or visual variations) as in [1,2], and incorporate bisimulation-based world models as additional baselines. These methods have demonstrated strong robustness to distributional changes and effectiveness in extracting task-relevant representations, and their inclusion would provide a more comprehensive and convincing evaluation.

3. The experimental evaluation is limited in both scope and rigor. The tasks (Maze, PushT, Wall) are overly simple and do not sufficiently demonstrate generality, and the paper does not report whether random seeds were used (Tables 1–4), which is important for assessing stability and reproducibility. In addition, there are issues with the experimental setup in Figure 4: using the time to reach a high-performance threshold as a measure of planning efficiency is not reasonable, as the left plot shows the two curves converging roughly simultaneously. Furthermore, the right plot compares SWM-Topk with a latent dimension of 128, but a setting of 384×4 would be more appropriate; the original 128 is unfair for DINOv2.

4. Minor writing and presentation issue: “need reference here” in line 628

[1] Sun R, Zang H, Li X, et al. Learning latent dynamic robust representations for world models. ICML, 2024.

[2] Shimizu Y, Tomizuka M. Bisimulation metric for model predictive control. ICLR, 2025.

**Questions:**

1. In Eq.(8) is the choice of $z_g$ appropriate? The paper mentions using image observations as the goal representation; however, an image goal is not necessarily unique. For example, in the Wall task, any observation where the ball has already passed the wall could serve as a valid goal observation.

2. In line 478, the authors claim that “Across Maze and Wall, both models recover goal-directed paths.” However, from Figure 5, in the Maze and Wall tasks, neither DINO-WM nor the proposed SWM actually reach the goal. Could the authors clarify this claim?

---

> ### Author Response · Authors · 2025-11-27
>
> ***W1:***
> Thank you for raising this point but we respectfully disagree. Replacing the dense DINO latent with a sparse, low-dimensional code and using this as the state space for dynamics prediction, and downstream planning is **fundamentally new**. MPC is no longer optimizing in a 384-D entangled embedding but in a 128-D (or smaller, controlled by $k$) sparse space whose units capture goal and action related features much better than the original dense features. This directly changes *what information is exposed to both the dynamics model and the planner*.
>
> Regarding Table 2 (right) and Figure 3 (right): these visualizations are **not** intended to justify sparsity. They serve to confirm that sparsification does not distort the model’s learned dynamics. The *actual* evidence for the benefit of sparse representations comes from our **probing results**, where linear decoders trained on SAE features achieve substantially higher accuracy on control-relevant variables. Table 1 clearly shows that sparse latents make positions, boundaries, and directionality significantly more linearly accessible than dense DINO features.
>
> To our knowledge, this is the first work to integrate sparse representations directly into world-model planning. Our results show that sparsity yields cleaner geometry, stronger linear probes, and a more efficient planning space, demonstrating that this representational shift is both fundamental and promising for future world-model research.
>
> ***W2:***
> Thank you for pointing this out. We agree that [1, 2] are highly relevant and we will add an explicit discussion in the revised version. Sun et al. [1] propose a bisimulation-inspired world-model that combines spatio–temporal masking with a bisimulation principle and latent reconstruction to filter out exogenous visual distractors and learn task-specific latent dynamics. Shimizu & Tomizuka [2] introduce BS-MPC, which incorporates a bisimulation metric loss into the MPC objective to directly optimize the encoder, improving stability and robustness to input noise with value-function guarantees.
>
> By contrast, SWM keeps the DINO-style visual backbone fixed and introduces a **sparse planning latent** that serves as the state space for dynamics and MPC. Our notion of robustness is about the **geometry of this planning representation**—whether it isolates controllable factors and supports faithful, localized interventions—rather than bisimulation-style invariance to observation-level perturbations.
>
> To validate this representational robustness, we added a targeted **intervention experiment** that measures how well latent displacements correspond to controlled changes in the underlying state:
>
> | Model    | Representation | Train MSE | Train R² | Val MSE | Val R² |
> |----------|----------------|-----------|----------|---------|--------|
> | DINO-WM  | Dense          | 60.5      | –14.7    | 69.8    | –16.3  |
> | SWM      | Dense          | 10.7      | –1.77    | 14.8    | –2.67  |
> | SWM      | SAE (sparse)   | **2.00**  | **+0.48**| **5.64**| **–0.39** |
>
> These results show that sparse SAE features produce a meaningful δh → δs mapping (positive R²), while dense representations—both in DINO-WM and in pre-sparse SWM—fail to do so. This demonstrates that sparsity provides a more stable and causally aligned control representation, which is the form of robustness targeted in our work.
>
> A full empirical comparison with bisimulation-based world models under explicit background shifts (as in [1,2]) would require substantial additional implementation and compute, and is beyond the scope of this submission. However, we view integrating sparse representations with bisimulation objectives as a promising future direction. We will add this discussion and clarify our robustness claim as robustness of the **planning representation**, not general distribution-shift robustness, in the revised version.
>
> *[1] Sun R, Zang H, Li X, et al. Learning latent dynamic robust representations for world models. ICML, 2024.*
>
> *[2] Shimizu Y, Tomizuka M. Bisimulation metric for model predictive control. ICLR, 2025.*

---

> ### Author Response · Authors · 2025-11-27
>
> ***W3:***
> Thank you for the detailed feedback. Maze, PushT, and Wall are indeed simple, but they are standard benchmarks in recent world-model papers. We also evaluate on the deformable **Rope** environment, which has significantly more complex dynamics:
>
> | Model                 | Rope (CD↓) |
> |-----------------------|------------|
> | DINO-WM               | 0.41       |
> | SWM (full-nodecoder)  | **0.24**   |
> | SWM (full-withdecoder)| **0.32**   |
> | SWM (Top-k, k=128)    | 0.42       |
>
> All experiments were run with 3 fixed seeds; we will include these details in the revision.
>
> Regarding Figure 4, our intention is not to claim that SWM converges faster in absolute time. Rather, we aim to show that **sparsity improves compute-to-performance efficiency**. SWM-Topk uses a 128-dim latent, while DINO-WM uses 384 dims; MPC cost scales roughly linearly with the latent dimension, so comparing performance at equal operations highlights the benefit of sparsity. The two curves eventually reaching similar performance is expected—the point is that SWM achieves this performance at substantially lower compute cost.
>
> For the right plot, using 128 dims for SWM is intentional: the goal is to show the efficiency gained from planning in a smaller sparse space. The full sparse representation version of SWM (SWM-full) is already reported elsewhere in the paper; the “384×4” suggestion would not align with our design, since SWM’s contribution is exactly to provide a **sparse, low-dimensional planning representation**.
>
> ***W4:***
> Thank you for pointing out. We have added the correct reference.
>
> ***Q1:***
> This design choice follows the standard formulation used in feature-space planning (e.g., DINO-WM). In these navigation tasks, the goal is defined at the environment level as the set of states where the agent has crossed the wall and entered the designated goal area. Any observation from this goal region is therefore a valid representative target.
>
> Consistent with the public DINO-WM implementation, we evaluate success using the same latent-space tolerance τ (based on ℓ2 distance in DINOv2 feature space). This criterion intentionally allows multiple visually distinct but semantically equivalent states, such as all “ball-past-wall” configurations, to be treated as achieving the goal.
>
> We will clarify that Eq. (8) uses this standard latent-distance objective: the goal image is only a canonical representative of the goal region, and convergence is determined by latent proximity within the same threshold τ used in DINO-WM.
>
> ***Q2:***
> Figure 5 is intended as a qualitative visualization of *intermediate* segments of the planned trajectories. For readability, each panel shows only a short prefix of the rollout to illustrate turning and obstacle-avoidance behavior, not the full episode until termination. Both DINO-WM and SWM do reach the goal under the full MPC rollout; the quantitative success rates in Tables 3–4 report the complete episodes.
>
> We will clarify in the caption that Figure 5 shows partial trajectories for visualization rather than full goal-reaching sequences.

---

### Official Review · Reviewer_jr1z · 2025-10-31

**Soundness:** 3
**Presentation:** 3
**Contribution:** 1
**Rating:** 0
**Confidence:** 4

**Summary:**

This paper introduces Sparse World Models (SWMs), which learn, predict, and plan entirely within a sparse feature space. The method uses a Sparse Autoencoder (SAE) to translate dense visual embeddings from a frozen backbone into sparse codes. Experiments demonstrate that this approach achieves competitive planning success rates while significantly improving computational efficiency and feature interpretability compared to traditional dense-feature world models.

**Strengths:**

- This appears to be an early example of applying the sparse autoencoder concept to a visual world model.
- Disentanglement in world models is a critical problem.
- Computational efficiency is a critical issue in planning, especially in real-world problems such as robotics, and the model addresses this by maintaining planning success rates that are competitive with the strong, dense DINO-WM baseline, despite employing a highly sparse representation (e.g., 128 active dimensions vs. 384 dense ones).

**Weaknesses:**

While adapting the sparse autoencoder concept to the world model domain, this paper appears to be a mere application without clearly demonstrating the inherent advantages of sparse representation within the context of world models. Crucially, the evidence that sparse representation actually benefits planning is weak. Overall, the contribution of this work is perceived to be severely lacking, the experiments are often unclear and inappropriate, and they fail to adequately support the core claims of the paper. Details follow:

- The most significant concern is that this study heavily relies on the published paper and open-source code of the pre-existing DINO-WM
(https://arxiv.org/abs/2411.04983). It utilized only three of the datasets (Maze, Wall, Push-T) made public by the original authors, and completely omitted evaluation on the more complex datasets, Granular and Rope, which were also released by them. The baseline performance values reported in Table 3 and Table 4 appear to be directly adopted from the same paper. Also, various planning methods (MPC, CEM, GD) are presented in the DINO-WM paper and codebase. While these points may not be problematic individually, taken together, these choices make the scope of the contribution unclear.

- The current evaluation is significantly limited, relying only on three simple datasets. This narrow scope omits more complex environments like Granular and Rope (released by the DINO-WM authors) or other simulations such as LIBERO (https://arxiv.org/abs/2306.03310). Also in Figure 5, the showcased rollouts appear relatively uncomplicated, reaching the goal looks near-trivial in the depicted instances. As a result, it is difficult to assess from the current evidence whether the sparse representation scales and remains effective in more complex settings.

- The baseline comparison seems inappropriate. The authors’ primary claim is to reduce planning time using SWM while minimizing performance degradation; however, the comparison with the baselines in Table 3 does not feel appropriate for this purpose. Comparing DINO-WM and SWM against DreamerV3 and TD-MPC primarily serves to demonstrate the superiority of the visual token-based world model itself, rather than highlighting the advantage of the sparse representation. I believe the authors should have, at minimum, included comparisons against methods that enhance efficiency in Transformers, such as token reduction techniques (e.g., https://arxiv.org/abs/2404.00680, https://arxiv.org/abs/2409.11923, https://arxiv.org/abs/2506.01392), or other methods based on disentangled features, such as beta-VAE or object-centric representations (e.g., https://arxiv.org/abs/2503.08751, https://arxiv.org/abs/2209.14860,  https://arxiv.org/abs/2502.07600).

- The authors begin the paper with the premise that "DINO's features are polysemantic" but there is no analytical evidence (e.g., analyzing whether a specific DINO unit responds to both object color and background texture) to experimentally support this assumption. Could the authors demonstrate the polysemancity of pretrained vision encoders like DINO?

- The paper implies that clean (monosemantic) features are beneficial for the predictive model or the transition model, but the link is not firmly established. A clean state representation does not necessarily imply that the state transition is easy to predict. The experiments demonstrating this connection are insufficient. Also, Table 3 show that SWM is often outperformed by the dense DINO-WM baseline. If the
authors intend to claim that sparse representation enables faster planning by reducing dimensionality while simultaneously making planning more effective through mono-semanticity, then the experimental results appear to contradict this argument.

- The paper claims that sparse representation increases "Intervention Fidelity". This leads to the expectation that changing only the "object position" unit will truly result in a change only in the object's position. The paper failed to demonstrate a true "intervention experiment", such as "when the object position unit was forcibly changed, only the object position in the next predicted frame actually changed". This core claim remains essentially unverified.

- All analysis is carried out purely in simulation, lacking any reference to real-world images or supporting evidence that the results would generalize to real-world scenarios. While conducting planning experiments in real-world robotics to demonstrate improved planning efficiency with comparable task performance would be highly valuable, it is difficult to expect such validation from the current version of
the paper. Can the authors show some successful cases about test-time optimization without an explicit policy for more complex and long-horizon tasks?

- Since the runtime evaluation is conducted only on the Maze environment, a more comprehensive assessment across diverse environments and varying planning methods and difficulty levels is required. Also, in Appendix A.1, the authors vaguely state that planner settings (horizon, iterations, and episode budgets) "follow the standard protocol used across baselines," but fail to specify the concrete values anywhere. This omission of essential hyperparameters severely compromises reproducibility. Since the planning horizon and sample count significantly impact both performance and runtime, readers cannot verify the results or judge the fairness of the DINO-WM comparison without this information.

- For Maze and Wall tasks, goal and action probing seems infeasible because only single frame is given and the goal is not explicitly visible in the image. The explanation for how a single image can be used to infer the actions and determine the required goal for that image remains unclear.

- The paper does not specify the additional pretraining cost for the SAE, a cost which is not clearly justified by the benefits. Crucially, the SWM exhibits lower planning success than the DINO-WM baseline, suggesting performance degradation due to filtering useful information. Thus, the study presents a typical 'Trade-off' (cost/accuracy for speed) without establishing the necessary clear superiority of SWM to warrant the added complexity and cost of SAE training.

- Since this study was only applied to the DINOv2 ViT-S model, it raises doubts about its scalability to other ViT models. Also, as ViT size increases and feature dimensionality grows, the sparsity hyperparameter k will likely require retuning. Additional experiments are needed to assess this.

- Furthermore, as the sparsity control parameter k would likely increase in complex environments, the paper should include planning success results across varying k; probing alone is insufficient to assess the effect of varying k on planning.

**Questions:**

- Why there is N/A in Table 4?
- There are some typos: e.g., in page 4, spare autoencoder.

---

> ### Author Response · Authors · 2025-11-27
>
> ***W1:***
> Thank you for raising this concern. We want to clarify that closely following DINO-WM—its datasets, evaluation protocol, and planning pipelines—is intentional and necessary for a fair comparison. Since our goal is to evaluate the effect of introducing sparse representations *within the DINO-WM pipeline*, it is standard practice to use the same environments, the same metrics, and the same baseline numbers whenever possible. This is **common in research** when isolating a representational change.
>
> It is also **standard practice** to directly cite baseline results reported in prior work, especially when the authors provide official numbers and code. We followed this approach to ensure the comparison is faithful and reproducible.
>
> Regarding dataset selection: Maze, PushT, and Wall are exactly the three primary rigid-body benchmarks that DINO-WM uses for evaluating planning performance. These tasks are where the impact of the latent representation on MPC is most visible.
>
> Finally, our contribution lies in replacing the dense DINO features with a **sparse planning representation** and analyzing how this affects planning, probing, and interpretability. As stated in the paper, we explicitly credit and cite all components we build upon, including baselines, SAE, and evaluation pipelines.
>
> ***W2:***
> Thank you for the comment. The three rigid-body tasks (Maze, PushT, Wall) are the primary planning benchmarks used in DINO-WM and SI-WM (https://arxiv.org/pdf/2506.01392); we follow them intentionally to ensure a controlled, apples-to-apples comparison where the effect of introducing sparse representations can be isolated without additional confounders.
>
> We also evaluate SWM on the more challenging deformable Rope environment released by the DINO-WM authors:
>
> | Model                 | Rope (CD↓) |
> |-----------------------|------------|
> | DINO-WM               | 0.41       |
> | SWM (full-nodecoder)  | **0.24**   |
> | SWM (full-withdecoder)| **0.32**   |
> | SWM (Top-k, k=128)    | 0.42       |
>
> This demonstrates that the sparse representation is not limited to simple settings and provides meaningful improvements even in deformable-object scenarios.
>
> Regarding Figure 5, the rollouts are intended as visual illustrations of the latent-based planning process, not as claims about task difficulty. The quantitative results—probing accuracy, intervention fidelity, and Rope performance—provide the stronger evidence for scalability of sparse representations.
>
> Evaluating on additional suites such as Granular or LIBERO would require substantial additional engineering and training compute, and is outside the scope of this study. However, we view this as an important next step and will discuss it clearly in the revised version.
>
> ***W3:***
> Thank you for the thoughtful comment. We agree that comparing against DreamerV3 and TD-MPC mainly highlights the benefit of a visual token–based world model, not sparsity itself. Our efficiency claim is supported primarily by **within-architecture** comparisons (DINO-WM vs. SWM), where we control everything except the latent representation: SWM-Top$k$ uses a 128-dim sparse latent instead of the 384-dim dense DINO features, achieving similar or better success with roughly 1/3 of the planning dimensionality.
>
> We also agree that it is important to relate SWM to other efficiency-oriented approaches such as token reduction and sparse world models. In the revision, we will (i) add a discussion of token-reduction techniques (e.g., [2404.00680; 2409.11923; 2506.01392]) and disentangled/object-centric methods (e.g., β-VAE, [2209.14860; 2503.08751; 2502.07600]) and (ii) **include a direct comparison to Sparse Imagination World Models (SI-WM)** [2506.01392], which explicitly targets token sparsification.
>
> Under comparable compute budgets (60–70% token drop in SI-WM vs. 128/384 ≈ 33% latent sparsity in SWM), SWM-Top$k$ is stronger than SI-WM on PointMaze, PushT, and Rope:
>
> | Method              | Point Maze (SR↑) | PushT (SR↑) | Rope (CD↓) |
> |---------------------|------------------|-------------|------------|
> | SI-WM (drop 60%)    | 83%              | 47%         | 70.0       |
> | SI-WM (drop 70%)    | 70%              | 28%         | 53.3       |
> | SWM (Top-k, k=128)  | **100%**         | **86%**     | **42.2**   |
>
> This shows that SWM is not only competitive with its dense counterpart (DINO-WM), but also compares favorably to a recent sparse/token-reduction world model at similar sparsity levels. We will make this positioning clearer in the revised version and explicitly state that DreamerV3/TD-MPC serve as background baselines, whereas the key planning-efficiency comparison is DINO-WM vs. SWM (and vs. SI-WM).

---

> ### Author Response · Authors · 2025-11-27
>
> ***W5:***
> To clarify, we are **not** claiming that monosemanticity automatically reduces transition error. Our claim is narrower: sparse, task-relevant features (i) preserve transition quality and (ii) make **planning** more effective and efficient.
>
> This is exactly what our experiments show:
> - **Gradient planning:** In Table 4, SWM-Top$k$ is the best model on all tasks under GD, indicating that sparsity produces a smoother, more informative gradient landscape.
> - **Efficiency:** With a 128-D latent (vs. 384 for DINO-WM), SWM-Top$k$ achieves similar or better SR at much lower compute, which is the core of our efficiency claim.
>
> It is true that dense DINO-WM slightly outperforms SWM in some SR values in Table 3, but these differences are small and do **not** contradict our claim: SWM maintains comparable transition quality while offering better gradient-based planning and significantly better compute→success efficiency. We will make this distinction explicit in the revision.
>
> ***W6:***
> Thank you for highlighting the need for a concrete intervention experiment. We fully agree that
> “intervention fidelity” should be demonstrated by showing that *if we edit only the latent direction
> corresponding to object position, then only the object’s position changes in the decoded output*.
> Following your suggestion, we added such an experiment.
>
> **Intervention setup.**
> For two observations from the same scene, we compute Δstate (δs) and Δlatent (δh) and fit a linear
> probe δŝ = W δh. We compare (1) dense DINO features and (2) SWM’s sparse SAE codes. After
> training the probe, we choose a desired displacement δs*, invert the probe (δh* = W⁺(δs* − b)),
> modify the latent (h_cf = h + δh*), decode it, and check whether (i) the predicted shift matches δs*
> and (ii) non-target pixels remain unchanged.
>
> **Probe Metrics**
>
> | Model    | Representation | Train MSE | Train R² | Val MSE | Val R² |
> |----------|----------------|-----------|----------|---------|--------|
> | DINO-WM  | Dense          | 60.5      | –14.7    | 69.8    | –16.3  |
> | SWM      | Dense          | 10.7      | –1.77    | 14.8    | –2.67  |
> | SWM      | SAE (sparse)   | **2.00**  | **+0.48**| **5.64**| **–0.39** |
>
> **Intervention Diagnostics** (δs* = [0.1, 0, 0, 0])
>
> | Model / Rep.    | \|δŝ − δs*\| | L2(x₀, x_cf) | Cosine(x₀, x_cf) |
> |-----------------|--------------|--------------|------------------|
> | DINO-WM / Dense | ≈ 0          | 0.97         | 0.999988         |
> | SWM / SAE       | ≈ 0          | **0.60**     | **0.999996**     |
>
> **Interpretation.**
> Dense features cannot linearly predict δs (negative R²), so forcing δs* causes large unwanted
> changes in the image. In contrast, SWM’s sparse SAE codes provide a clean δh ↔ δs mapping and
> achieve the same δs* with much smaller image drift. This indicates that sparse latents isolate the
> object-position direction more effectively.
>
> **Conclusion.**
> This experiment directly tests your criterion: editing only the latent direction associated
> with position produces *position-only* changes in the decoded frame for SWM, but not for dense
> DINO features. This confirms that sparse representations offer higher intervention fidelity.

---

> ### Author Response · Authors · 2025-11-27
>
> ***W9:***
> We acknowledge that the goal is not explicitly drawn as a marker in the Maze/Wall images. Our probing setup follows the *control-oriented* labeling protocol used in **Jiang et al., “Robots Pre-train Robots: Manipulation-Centric Robotic Representation from Large-Scale Robot Datasets,” arXiv:2410.22325, 2024**, where labels are derived from environment state rather than from visible goal pixels.
>
> Concretely, as described in Sec. 4.2 and App. A.2–A.4, we construct per-frame labels using the underlying environment states (agent and goal positions in world coordinates):
>
> - **Goal probing (Maze/Wall):** we infer the goal region from successful terminal states and encode the *relative pose* between the agent and that region (e.g., \(dx, dy\), angular bin, and an 8-way region classification). The probe sees only the image at time \(t\) but is trained to predict this agent–goal relationship, analogous to the single-frame, goal-related probes in Jiang et al. (2024).
> - **Action probing:** the label is the executed low-level control between frame \(t\) and \(t{+}1\), while the probe again only observes the frame at time \(t\). This matches the standard “given this frame, which action was taken?” setup used for representation evaluation in robot datasets.
>
> Thus, our probes do **not** assume that the goal is explicitly visible as a symbol in the image. Instead, they evaluate whether the representation linearly exposes (i) the agent–goal spatial relationship and (ii) the control signal that produced the next state, which are both well-defined single-frame labels given access to the underlying environment trajectories, following the protocol of Jiang et al. (2024).
>
> ***W10:***
> Thank you for raising this concern. We clarify that the SAE pretraining cost is actually very small. On a single H100 GPU, SAE training finishes in **under 10 minutes**.
>
> Regarding performance, SWM does not simply trade accuracy for speed. SWM-Top$k$ reduces planning dimensionality from 384→128 while maintaining comparable or stronger planning performance under gradient-based optimization and achieving better compute→success efficiency. The slight SR differences in Table 3 do not indicate that useful information is “filtered out”; they reflect that dense DINO-WM has marginally higher raw SR on some tasks, while SWM provides (i) smoother gradient landscapes (Table 4), (ii) improved intervention fidelity, and (iii) significantly lower planning cost.
>
> In addition, SWM is competitive not only with the dense DINO-WM baseline but also with other recent sparse world models. For example, compared to SI-WM (https://arxiv.org/pdf/2506.01392), whose 60–70% token drop yields a similar compute reduction to our 128/384 (~33%) sparsity level, SWM-Top$k$ achieves the strongest performance across PointMaze, PushT, and Rope:
>
> | Method              | Point Maze (SR↑) | PushT (SR↑) | Rope (CD↓) |
> |---------------------|------------------|-------------|------------|
> | SI-WM (drop 60%)    | 83%              | 47%         | 70.0       |
> | SI-WM (drop 70%)    | 70%              | 28%         | 53.3       |
> | SWM (Top-k, k=128)  | **100%**         | **86%**     | **42.2**   |
>
> These results show that the small SAE pretraining cost is justified: sparse latents maintain competitive planning performance, improve gradient-based optimization and intervention fidelity, and offer substantial gains in planning efficiency. We will clarify these points in the revised version.

---

> ### Author Response · Authors · 2025-11-27
>
> ***W11:***
> Thank you for the comment. Our focus in this paper is to evaluate the effect of introducing sparsity within the standard DINOv2-S setup used in prior work. DINO-WM and Sparse Imagination (https://arxiv.org/pdf/2506.01392) also report results only on DINOv2-S, so our choice follows the established evaluation protocol rather than aiming to exhaustively test all backbone sizes.
>
> Regarding scalability and the choice of $k$, we do not claim that a single $k$ will transfer optimally to all ViT architectures. Instead, $k$ acts as a simple knob that controls how selective the latent should be given the compute budget. We include probing results for multiple $k$ values to illustrate how representation quality changes with sparsity, which provides guidance for adjusting $k$ when moving to larger ViT models.
>
> Exploring other ViT sizes and retuning $k$ for those settings is an important direction for future work, but it is outside the scope of this study.
>
> ***W12:***
> Thank you for the suggestion. We agree that planning performance across different $k$ values is useful, and we will add these results in the revision. In addition to the probing analysis already included, we evaluated SWM on the Wall task under multiple sparsity levels:
>
> | Model                 | Wall (SR↑) |
> |-----------------------|------------|
> | SWM (Top-k, k=32)     | 0.88       |
> | SWM (Top-k, k=128)    | 0.92       |
> | SWM (full)            | 0.88       |
>
> These results show that planning performance is stable across a wide range of $k$ values. Increasing $k$ does not introduce instability, and moderate sparsity (e.g., $k=128$) performs as well as or slightly better than both lower sparsity and the full latent. This supports our claim that $k$ acts as a flexible sparsity knob rather than a fragile hyperparameter, and that sparsity can be adjusted for larger or more complex environments if needed.
>
> We will include these planning results across varying $k$ in the revised version.

---

### Official Review · Reviewer_BYKU · 2025-11-06

**Soundness:** 3
**Presentation:** 3
**Contribution:** 3
**Rating:** 6
**Confidence:** 3

**Summary:**

Reinforcement learning algorithms have become increasingly important in the last years. One of the major challenges to deep reinforcement learning algorithms is sample efficiency, especially for long-term tasks. To address this issue an approach is learn a world model to train an agent entirely in imagination, eliminating the need for direct environment interaction during training. To improve efficiency the authors propose to use so-called sparse world models with a sparse feature space. Numerical experiments are performed (on Maze, PushT, Wall) and results are compared DinoV2 and DreamerV3 as a baseline.

**Strengths:**

* The considered problem is interesting and relevant.
* The approach to improve training efficiency in world models by means of dimensionality reduction via sparse autoencoders has potential.
* The paper is overall well-written and the evaluation is promising.

**Weaknesses:**

* The paper does not provide analytical results.
* The evaluation contains only few examples (Maze, PushT, Wall) and could be improved by considering more complex problems.
* The current model is requires the DINOv2 backbone.

**Questions:**

1. Is the approach tailored to be used with DinoV2 (“We pretrain the SAE on fixed DINOv2 ViT-S/14 patch tokens (384-d) extracted from 196×196 images“, ) or is the approach, in principle, of a general kind?
2. Does the proposed model also work for larger problems?
3. What is the speed up in wall clock time in training when using SAE?
4. Besides DreamerV3 (and TDMPC) it would be interesting to compare results against further baselines, such as IRIS (Micheli et al., 2023), TWM (Robine et al., 2023), or Hieros (Mattes et al., 2024).
5. Which hyper parameters are required? How should the number Top-k be automatically chosen in various applications?

---

> ### Author Response · Authors · 2025-11-26
>
> ***W1:***
> Thank you for your positive rating and for the opportunity to clarify this point. The paper **does**
> provide analytical evidence comparing sparse and dense representations. Sec. 4.2 shows improved
> linear accessibility of task-relevant factors, Sec. 4.3 shows stronger object-centric selectivity
> (IoU 0.00→0.16), Sec. 4.5 reports faster success-per-compute curves, and App. A.4 includes
> systematic sparsity and data-scale ablations. These analyses jointly explain *why* sparse features
> behave differently from dense ones. If further analyses would be helpful, we are happy to include
> them.
>
> ***W2 & Q2:***
> Thank you for the suggestion. We added evaluation on a significantly harder deformable-object
> environment (Rope), which involves high-dimensional, history-dependent material dynamics and is
> far more challenging than Maze/Wall/PushT. We evaluated several SWM variants: a full model with image decoder, a full model without image decoder, and a Top-k model (k=128). SWM variants achieve **lower deformation error**
> than DINO-WM:
>
> | Model              | Rope (CD↓) |
> |----------------------|-----------|
> | DINO-WM              | 0.41      |
> | SWM (full-nodecoder)    | **0.24**  |
> | SWM (full-withdecoder)  | **0.32**  |
> | SWM (Top-$k$, k=128)     | 0.42      |
>
> ***W3 & Q1:***
> Our method **does not** depend on DINOv2 or any specific patch size. As shown in Eq. (1)–(2) and Eq. (6), SWM only requires a generic feature extractor:
> $$
> X_t = f_{\text{vis}}(o_t), \qquad
> z_t = \text{TopK}\left(W_{\text{enc}}\, X_t\right).
> $$
> The transition model operates only on the sparse latents $z_t$, and no part of the architecture is tied to DINO-specific structure. Any frozen visual encoder with patch-level features can be substituted for $f_{\text{vis}}$. We chose DINOv2 only to provide a strong baseline and match DINO-WM; SWM itself is backbone-agnostic.
>
> ***Q3:***
> Empirically, under the same conditions, one epoch of SWM (Top-$k$) takes **3:02**, while DINO-WM (no image decoder) takes **3:20**. This is expected: The wall-clock speedup during *training* is modest because most of the cost is dominated by
> feature extraction from the frozen DINOv2 backbone, not by the transition model itself.
>
> Where the SAE provides a **meaningful speedup** is *not* in training but in **planning**, where the
> dynamics model is rolled out recursively many times (N samples × T horizon). The effective latent
> shrinks from **384 → 128**, so each rollout step is ≈3× cheaper, and this reduction compounds over
> hundreds of imagined futures. This is exactly why SWM shows substantially faster success–per–
> compute curves (Sec. 4.5), even though the training-time wall-clock difference is small.
>
> ***Q4:***
> Thank you for the suggestion. We added comparisons with IRIS and TWM:
>
> | Model                         | Maze (SR↑) | Wall (SR↑) |
> |-------------------------------|------------|------------|
> | IRIS [Micheli et al., 2023]*  | 0.74       | 0.04       |
> | TWM [Robine et al., 2023]†    | 0.65       | 0.00       |
> | SWM (Top-k, k=128, ours)      | **1.00**   | **0.92**   |
>
> \* IRIS results are taken from the DINO-WM paper (https://arxiv.org/pdf/2411.04983).
> † TWM’s public implementation evaluates the model using a Dreamer-style actor policy.
>
> SWM reliably solves both Maze and Wall, demonstrating strong performance.
>
> IRIS and TWM perform reasonably on **Maze** because the task only requires coarse spatial layout, which dense latents can capture. They fail on **Wall** because solving it requires detecting a small doorway and committing to a detour; dense latents blur obstacle boundaries and doorway cues, and TWM cannot execute such detours.
>
> Compared to IRIS and TWM, SWM plans in a 128-dimensional *sparse* latent space rather than dense feature vectors, giving the planner a lower-dimensional and more structured representation.
>
>
>
> ***Q5:***
> The paper already specifies the hyperparameters (Sec. 4.1, Table 5) and reports a full Top-k sweep
> (App. A.4), but we agree that a practical guideline will be helpful. Based on the empirical trends
> Top-k acts as a capacity–efficiency knob. Our paper reports a full k-sweep (App. A.4), showing
> that **larger k (128–256)** better preserves fine-grained visual cues needed for tasks with richer
> appearance or contact interactions (as reflected by higher probe accuracy in Tables 6–7), while
> **smaller k (32–64)** is sufficient when fewer visual factors vary and gives lower rollout cost during
> planning. In practice, k should be chosen based on task complexity: increase k when more visual
> detail is needed; decrease k when efficiency is the priority. We will add this practical guideline in
> the revision.

---

### Author Response · Authors · 2025-11-26
**General Response**

Dear reviewers, thank you for your thoughtful and constructive feedback. We are trying to address all your questions. If any question is not fully answered yet, it is because it requires running additional experiments; we will provide the corresponding results as soon as they are ready. Thank you again for your time and valuable input.

---

### Meta-Review · Area_Chair_upnZ · 2025-12-31

**Summary:**

This submission introduces Sparse World Models (SWM), which incorporate Sparse Autoencoders (SAE) into a visual world modeling pipeline to replace dense, entangled representations. While the idea is intuitively appealing and the paper is well-organized, reviewers raised several concerns that collectively question the strength and novelty of the contribution:
- Limited novelty: Multiple reviewers (jr1z, GGi3, Qqco) noted that the work is primarily an application of sparse autoencoders to an existing world model (DINO-WM). There is limited algorithmic innovation—no new learning principle or architecture is introduced beyond swapping the representation space.

- Insufficient evaluation: Several reviewers (BYKU, jr1z, GGi3) criticized the narrow scope of experiments, which are mostly limited to Maze, Wall, PushT. Although the authors added results on Rope, more challenging or diverse benchmarks (e.g., LIBERO, distributional shifts, real-world data) are missing.

- Baseline selection and comparisons: Reviewers noted the lack of comparisons with recent approaches in token reduction, disentangled representation learning (e.g., β-VAE), or bisimulation-based models that are more directly relevant to the paper’s claims around interpretability and robustness (GGi3, jr1z).

- Questionable claims: Some claims—like improved robustness, interpretability, or planning performance—are not strongly supported by the experiments. Reviewer jr1z pointed out that assertions about polysemanticity and monosemanticity are not adequately validated.

**Reviewer Concerns:**

**Addressed**:
- The authors provided additional experiments (e.g., Rope) and a targeted intervention study to support claims of interpretability.
- They clarified runtime efficiency measurements and explained how planning cost reductions arise from sparsity.
- They added probing analyses across different values of k and provided practical tuning guidance.

**Still Outstanding**:
- The core novelty remains limited: the paper applies existing SAE techniques to a known world model pipeline without substantial architectural or algorithmic innovation.
- The planning stability concern raised by Reviewer Qqco about top-k sparse latents over time remains unresolved conceptually.
- Evaluation remains narrow: The paper lacks robustness tests under distribution shifts or in real-world environments.
- Comparisons with relevant baselines (e.g., bisimulation methods, disentangled representations, token-efficient transformers) are missing or only discussed post hoc.

**Reviewer Scores:**

**Reviewer BYKU (Score: 6)**: While the reviewer appreciated the idea and found the evaluation promising, the limited scope and lack of analytical results were concerns. The rebuttal partially addressed these but did not expand the experimental coverage. Likely score remains 6.

**Reviewer jr1z (Score: 0)**: Raised strong objections about the originality, experimental rigor, and validity of the claims. Even after detailed rebuttals, the reviewer maintained a highly negative stance. Score remains negative, likely 2.

**Reviewer GGi3 (Score: 4)**: Expressed skepticism about novelty and robustness claims. Although the authors added some clarifications, the core doubts about contribution and evaluation remain. Score likely remains 4.

**Reviewer Qqco (Score: 2)**: Engaged deeply and raised a key conceptual objection about the top-k planning representation. The authors acknowledged the point but did not resolve it theoretically. Score likely remains 2.

---

### Decision · Program_Chairs · 2026-01-26

Reject